# Sustainable Exploitation of Apple By-Products: A Retrospective Analysis of Pilot-Scale Extraction Tests Using Hydrodynamic Cavitation

**DOI:** 10.3390/foods14111915

**Published:** 2025-05-28

**Authors:** Luca Tagliavento, Tiziana Nardin, Jasmine Chini, Nicola Vighi, Luca Lovatti, Lara Testai, Francesco Meneguzzo, Roberto Larcher, Federica Zabini

**Affiliations:** 1HyRes S.r.l., Via Salvator Rosa 18, 82100 Benevento, Italy; luca.tagliavento@hyres.it; 2Technological Transfer Center, FEM-IASMA Fondazione Edmund Mach, Istituto Agrario di San Michele all’Adige, Via E. Mach 1, 38010 San Michele all’Adige, Italy; tiziana.nardin@fmach.it (T.N.); roberto.larcher@fmach.it (R.L.); 3Consorzio Melinda SCA, Via Trento 200/9, 38023 Cles, Italy; jasmine.chini@melinda.it (J.C.); nicola.vighi@melinda.it (N.V.); 4CIF Consorzio Innovazione Frutta, Via Edmondo Mach 1, 38010 San Michele all’Adige, Italy; lovatti@cif.tn.it; 5Department of Pharmacy, University of Pisa, 56126 Pisa, Italy; lara.testai@unipi.it; 6Institute of Bioeconomy, National Research Council of Italy, Via Madonna del Piano 10, 50019 Florence, Italy; federica.zabini@cnr.it

**Keywords:** apple by-products, bioactive compounds, bioeconomy, green extraction, hydrodynamic cavitation, sustainability

## Abstract

Apple by-products (APs) consist of whole defective fruits discarded from the market and pomace resulting from juice squeezing and puree production, which are currently underutilized or disposed of due to the lack of effective and scalable extraction methods. Bioactive compounds in APs, especially phlorizin, which is practically exclusive to the apple tree, are endowed with preventive and therapeutic potential concerning chronic diseases such as cardiovascular diseases, metabolic diseases, and specific types of cancer. This study investigated the exploitation of APs using hydrodynamic cavitation (HC) for the extraction step and water as the only solvent. High-temperature extraction (>80 °C) was needed to inactivate the polyphenol oxidase; a strict range of the cavitation number (around 0.07) was identified for extraction optimization; less than 20 min were sufficient for the extraction of macro- and micro-nutrients up to nearly their potential level, irrespective of the concentration of fresh biomass up to 50% of the water mass. The energy required to produce 30 to 100 g of dry extract containing 100 mg of phlorizin was predicted at around or less than 1 kWh, with HC contributing for less than 2.5% to the overall energy balance due to the efficient extraction process.

## 1. Introduction

Apples, a very popular fruit, are consumed all over the world in a variety of products including fresh fruit, juice, cider, concentrate, and puree. They are rich in valuable chemical compounds (e.g., phenolic compounds, pectin, and fibers), in variable amounts depending on cultivar, growing, and harvest conditions, and degree of ripeness [1].

To date, more than 60 phenolic compounds have been found in apples, namely phenolic acids (hydroxycinnamic and hydroxybenzoic acids), flavanols (catechin, epicatechin, procyanidin B2), flavonols (quercetin, isoquercetin), dihydrochalcones (phloretin and its glucoside phlorizin), and anthocyanins [2,3], differently distributed in peels, flesh, and seeds. These phenolic compounds make apples important sources of natural antioxidants. Epidemiological studies have shown that frequent apple consumption results in reduced risk of chronic pathologies such as cardiovascular disease, specific cancers, and obesity [4,5,6].

While most apple fruits are aimed at direct consumption, about 11.6 Mtons/year of fresh fruit are processed to obtain apple products, leading to about 3.5 Mtons/year of by-products also known as apple pomace (AP), which is usually discarded or used for animal feeding [7]. AP is composed of peels and flesh (95%), seeds (2–4%), and stem (1%), and its dietary fiber content consists of pectin, cellulose, hemicellulose, and lignin [8].

The proximate and analytical compositions of AP depend on the variety and origin of apples, as well as on the pre- or post-harvest interventions, such as pressing and other processes to obtain juice, cider, wine, and distilled spirits [9]. Based on published data, total sugars compose up to 50% of the dry matter, total dietary fiber around 27%, fat and protein up to 4% each, while the total phenolic content amounts to about 300 mg GAE/100 g [10]. Dominant phenolics in AP are similar to the whole fruit [11]. Due to the lower water content compared to the whole fruit and the higher content of phenolic compounds, AP represents an interesting low-cost source of phytochemicals and bioactive compounds [3,12].

Polyphenols are extensively recognized for their capacity to enhance both antioxidant defenses and the inflammation response in humans. The antioxidant activity of apple-derived polyphenols is primarily attributed to their ability to scavenge reactive oxygen species and reactive nitrogen species, thereby neutralizing free radicals. This mechanism involves the interaction of hydroxyl groups with the p-electrons of the aromatic benzene ring, facilitating electron transfer and radical stabilization [12]. The incorporation of antioxidant-rich apple by-products into the diet may help strengthen the cellular antioxidant defense system, thereby mitigating oxidative stress, which is a key etiological factor in the pathogenesis of various chronic diseases including diabetes, obesity, cardiovascular diseases, and cancer [4,13]. Recent studies have further elucidated the ability of polyphenols to activate the nuclear factor erythroid 2 (NRF2) signaling pathway. NRF2 is a transcriptional factor that regulates the expression of a wide array of antioxidant enzymes, which are involved in the regulation of cellular redox homeostasis and inflammation. This pathway plays a pivotal role in the cellular defense against oxidative and inflammation-mediated damage by promoting the transcription of genes involved in antioxidative and anti-inflammatory responses [14,15,16].

Several studies have also highlighted the role of polyphenols in positively influencing blood lipid profiles mainly due to their remarkable antioxidant properties. The polyphenol-rich fraction of apples, including procyanidins, hydroxycinnamic acids, and flavonols, has been suggested to play a key role in reducing total cholesterol and LDL cholesterol levels [17,18].

Growing evidence underscores the beneficial effects of apple phenolic compounds on the digestive tract [19,20]. These compounds are also valuable for their antimicrobial properties, primarily attributed to the activities of phloretin and phlorizin [21,22].

Of note, phlorizin, and in lower amounts its aglycone phloretin, are present in different parts of the apple tree, among which the fruits [23]. Phlorizin has been shown to display numerous pharmacological effects, of which antidiabetic properties are the most studied and it may be considered as a pioneer of the modern antidiabetic classes called gliflozins [24]. Indeed, it is a potent inhibitor of the sodium–glucose cotransporter-2 (SGLT2, in the nanomolar range), responsible for glucose reuptake. Though at lower efficacy, it also inhibits SGLT1 (expressed at the brush border membrane of enterocytes), therefore, usually it is defined as a dual SGLT1/2 inhibitor [25].

Besides phenolic compounds, fibers, carbohydrates, proteins, amino acids, fatty acids, minerals, and vitamins contained in AP are also gaining attention due to their health-promoting values. AP polysaccharides have been investigated for their hepatoprotective, antioxidant, and anticarcinogenic effects [26], and have been successfully applied both in the food and pharmaceutical industries [27]. Compared to the administration of the individual components, the concomitant administration of pectin and polyphenols has been ascribed superior efficacy in managing metabolic disorders, such as dyslipidemia and insulin resistance, improving lipid metabolism and reducing systemic inflammation, which collectively contribute to improved metabolic and cardiovascular outcomes [17,18,28].

AP remains significantly underexploited, especially considering its potential as a source of valuable molecules with a wide range of application areas, from food to pharmaceutical and nutraceutical industry. Conventional extraction techniques of phenolic compounds from whole apples and AP include Soxhlet using ethanol as a solvent and maceration with pressurized hot water. Soxhlet extraction still represents the standard method for solid–liquid extraction and has shown efficacy in the isolation of compounds like phlorizin, epicatechin, quercetin, and phloretin from AP with a yield of phenolic compounds equal to 4.13 mg/g [29]. However, this method presents several drawbacks, including long process time, large solvent volumes and wastewater and toxic solid waste streams [30].

New extraction technologies such as microwave-assisted extraction, ultrasound-assisted extraction, supercritical fluid extraction, and pulsed electric field extraction, have been tested to improve the efficiency of the extraction process while reducing the use of chemicals and energy consumption and generating safe, high-quality extract for various industrial applications [31]. To the best knowledge of the authors, none of such techniques has yet been proven at the pilot scale. Only one publication reported on the use of hydrodynamic cavitation (HC), an emerging, green, and scalable extraction technique, in a pilot-scale extraction of pre-fermented AP [32]. However, a stator-rotor setup was used, which jeopardizes full scale applications due to the excessive energy cost of rotational HC reactors compared to static ones, such as Venturi or orifice constrictions [33].

Cavitation in liquid media is a complex multiphase phenomenon involving the formation, expansion, and near-adiabatic collapse of vapor-filled bubbles within an oscillating pressure field. This process generates intense pressure shockwaves (reaching up to 1000 bar), high-speed hydraulic jets, localized extreme temperatures (up to thousands of Kelvin), and the production of free radicals, particularly hydroxyl groups [34,35].

Among various cavitation technologies, HC stands out as the only fully scalable solution. HC can be achieved by passing a liquid or a liquid–solid mixture through static constrictions of different geometries or by using specialized immersed rotary equipment. HC methods have shown great effectiveness and efficiency in applications such as food processing, process intensification, and the extraction of natural products, among numerous other uses [36,37].

With static HC reactors, the simplest representation of cavitation regimes is given by the cavitation number (σ), derived from Bernoulli’s law and shown in Equation (1):(1)$$\sigma =\frac{p_{2}-p_{sat}}{0.5\rho u^{2}},$$
where *p*_2_ is the recovery pressure downstream the throat (Pa); *p*_sat_ is the saturation vapor pressure of the liquid (Pa); *ρ* is the liquid density (kg·m^−3^); and *u* is the flow velocity through the throat (m·s^−1^) [38].

The intensity of cavitation escalates as the cavitation number decreases, up to the point of choked cavitation, where a significant increase in the number of cavities occurs, filling the downstream region of the reactor, coalescing and damping the energy release [39]. In distilled water, developed cavitation is observed within the range of 0.1 < σ < 1 [40]. Cavitation can also manifest around the impeller of a centrifugal pump, characterized by the standard cavitation number as defined in Equation (1) [41], where the velocity term *u* represents the peripheral velocity of the impeller. For most HC processes conducted under atmospheric pressure, the recovery pressure term *p*_2_ can be approximated to the atmospheric pressure (around 1 bar at sea level) in both the throat and pump impeller cavitation zones [42].

Static HC reactors, such as those employing Venturi or orifice constrictions, have been shown to surpass rotary reactors, particularly in large-scale applications [33,43], as they require lower pressure and energy inputs to achieve the same flow velocity [44]. In static reactors, the cavitation number σ can be easily regulated by adjusting the flow velocity u, either by modifying the reactor’s geometry or altering the pump frequency used to circulate the liquid or mixture. Additionally, all other factors being constant, σ is also influenced by the temperature of the circulating medium, due to the temperature-dependent properties of *p*_sat_ and *ρ*.

HC-based methods similar to the one used in this study were applied to the extraction of plenty of natural products, among which fruits and fruit by-products, such as citrus [45] and pomegranate [46]. Such applications showed higher efficiency compared to other methods and higher in vivo functionality compared to reference products.

This study aimed to retrospectively analyze and discuss the results of recent pilot-scale experiments using HC with a linear, Venturi-shaped reactor, which were carried out on untreated AP for the general purpose of checking the feasibility and sustainability of the exploitation of apple by-products. These tests were conducted following initial trials aimed to produce food fortification ingredients, as reported elsewhere [27,47]. The choice of the Renetta apple variety was based on the substantially higher content of bioactive compounds compared to other varieties cultivated in the same geographic area [48], as well as on a randomized, controlled, crossover clinical trial, showing that the regular and sustained consumption of two whole fruits per day provided beneficial hypocholesterolemic and vascular effects to healthy mildly hypercholesterolemic volunteers [49].

## 2. Materials and Methods

### 2.1. Raw Materials

Four batches of whole apple fruits of the Renetta variety were supplied by Consorzio Melinda S.c.a. (Cles, Trento, Italy) between November 2023 and January 2024. Each batch was stored in the dark at 4 °C for 24 h before processing. The fruits were crushed with a fruit mill (model MLP0002, Polsinelli Enologia Srl, Frosinone, Italy) to reduce the linear size to maximum 10 mm. Hereinafter, the apple raw material will be called apple pomace (AP). For three batches, the crushed fruits were squeezed using an hydropress (model Hydro 80 L stainless steel, Zambelli Enotech, Vicenza, Italy) until the amount of discarded juice was at least 50% of the original weight, to obtain a pomace that emulated the by-product of industrial juice squeezing or puree manufacturing.

### 2.2. Production of Apple Extracts

Three different custom-built HC devices, HC200, HC50 and HC300, with volume capacity of 200, 50 and 300 L, respectively, were used to obtain the apple extracts (AEs). Each device consisted of a closed hydraulic circuit and a circular Venturi-shaped reactor as the key components, where the liquid–solid mixture was inserted and moved by a centrifugal pump. In HC200, the pump was model ESHE 50-160/75 (Xylem Water Solutions Italia S.r.l., Lainate, Milan, Italy), nominal power 7.5 kW, open impeller with a diameter of 174 mm, fixed frequency of 50 Hz and rotation speed of 2900 rpm; in HC50, the pump was model HDM 25-19A (Salvatore Robuschi e C. S.r.l., Parma, Italy), nominal power 3.0 kW, open impeller with a diameter of 185 mm, adjustable frequency between 40 Hz (2320 rpm) and 60 Hz (3480 rpm) using the inverter; in HC300, the pump was model RDM 50-20BR 2C15 M126 3 (Salvatore Robuschi e C. S.r.l., Parma, Italy), nominal power 15.0 kW, open impeller with a diameter of 195 mm, adjustable frequency between 41 Hz (2400 rpm) and 60 Hz (3516 rpm) using the inverter. All the parts in contact with the circulating mixture were made of food-grade AISI 316 stainless steel.

Electricity was the only energy source. Devices HC50 and HC300 were equipped with an inverter ATV320U40N4C (Schneider Electric S.p.A., Stezzano, Italy) and AC Drive GA500 400V Class Three-Phase Input (Yaskawa, Orbassano, Italy) to tune the pump frequency. Device HC200 was further described in previous studies [50]. The processes were carried out at atmospheric pressure. When needed, the temperature rise was controlled using tap water circulating in channels around the outer wall of the device’s tank. Power and energy consumption were measured using three-phase digital power meters with power resolution 1 W and energy resolution 10 Wh: for devices HC200 and HC50, model D4-Pd (IME, Milan, Italy); for device HC300, model RTD100 OHM (SMC S.r.l., Collecchio, Italy). Figure 1 shows a schematic representation of the cavitation devices and the Venturi-shaped reactors.

Table 1 shows the basic features of the extraction tests: date, raw biomass, used HC device, fresh and dry apple biomass, solid to liquid ratio, process time, and temperature. AP was pitched in the extraction system at the beginning of each process, except for REP3 where four batches of AP were successively pitched. No additives were used to correct the pH level during the extraction processes, which was 3 ± 0.2. Immediately after the completion of the extraction process, the mixture was fed via a peristaltic pump (model AS50 3PH, Mori Luigi S.r.l., San Casciano In Val Di Pesa, Italy) to a 5 µm bag filter (model FBF-0102-AD10-050B, Findex Filtration S.r.l., Nerviano, Milan, Italy). Filtered samples were collected, stabilized using sodium fluoride (NaF), and rapidly cooled to −80 °C for subsequent analyses. Each experiment was replicated three times.

The experiments turned out to be sufficiently representative, as REP1 and REP2 explored nearly isothermal runs at different temperatures under otherwise similar conditions, while REP3 and REP4 explored a scale-up from REP2 at the upper temperature, as well as REW1, which used whole apples in place of apple pomace. However, it should be noted that this was a retrospective study of experiments originally aimed at investigating the general performance of the HC-based processing of the considered biomass.

### 2.3. Analysis of Raw Material and Extracts

Both AP and sampled apple pomace extracts (APEs) were analyzed to determine the total sugars content, the Oxygen Radical Absorbance Capacity (ORAC), the total phenolic content (TPC), the individual phenolic compounds, and the total dissolved solids (TDS), representative of the dry final products obtainable through the extraction processes, and, according to the following methods:Sugars: Glucose, fructose, sucrose, xylose and sorbitol were quantified according to the method developed by Di Lella et al. [51] using commercial standards provided by Merck (Darmstadt, Germany) 1 g of the raw apple material and 1 g of the extracted sample were dissolved in 40 mL of water, centrifuged, and the supernatant was subsequently diluted 25-fold for the raw material and 5-fold for the extracted samples. Quantification was performed using an ICS 5000 ion chromatograph (Dionex, Thermo Fisher Scientific, Waltham, MA, USA) equipped with a pulsed amperometric detector (PAD) consisting of a gold working electrode and a palladium reference electrode. The sugar content was calculated by summing the individual sugars. Linearity of the sugars was confirmed between 0.02 and 20 mg/L, and R^2^ were always >0.99. Repeatability (calculated as RSD% on three replicates) of 5% and uncertainty (σ/√2) of 4%.ORAC: Oxygen Radical Absorbance capacity was evaluated in according to Ou et al. [52], by dissolving 1 g of samples in 50 mL of an acetone:water mixture (50:50, *v*/*v*) for raw materials and 5 mL for extracted samples and appropriately diluting them with 10 mM potassium phosphate buffer (pH 7.4) for analysis. Subsequently, 150 µL of fluorescein working solution (1.2 µM) was added to microplate wells along with 50 µL of diluted buffer, standard (Trolox, 100 µM, Merck), control, and samples. The kinetic reaction with AAPH (2,2′-Azobis(2-methylpropionamidine) dihydrochloride) solution (41 g/L) took place in a fluorescence microplate reader (Varioskan Lux, Thermo Fisher Scientific, Waltham, MA, USA) and was measured every minute for 35 min (excitation at 485 nm and emission at 530 nm). Repeatability (calculated as RSD% on three replicates) of 11% and uncertainty (σ/√2) of 7%.TPC: Total polyphenol content was quantified adapting the protocol elaborated by Ceci et al. [53]. 10 g of the raw material were extracted with 40 mL of a water:methanol mixture (80:20, *v*/*v*) acidified with 0.85% H_3_PO_4_. The mixture was shaken for 15 min and centrifuged at 4 °C and 4000 rpm for 5 min (Rotina 380, Hettich, Germany). The supernatant was collected and stored at −20 °C until analysis. Extracts were diluted 25 times with the same solvent mixture. TPC was determined using the Folin–Ciocalteu method. 2 mL of the extract was added to 1 mL of Folin–Ciocalteu reagent, and the mixture was incubated for 5 min. Then, 5 mL of sodium carbonate solution (20% *w*/*v*) was added. After 90 min, the absorbance was recorded at 740 nm using a Cary 60 UV–Vis spectrophotometer (Agilent Technologies, Palo Alto, CA, USA) and compared to a standard curve of catechin [54]. Repeatability (calculated as RSD% on three replicates) of 16% and uncertainty (σ/√2) of 11%. TPC was expressed in (+)-catechin equivalent as recently used for the antioxidant capacity of flesh and peel of several apple cultivars [55].Phenolic profile: Individual phenolic compounds were quantified with a liquid chromatograph coupled to a heated electrospray ionization source (HESI-II) and a high-resolution Q-Exactive™ hybrid mass spectrometer (HPLC-HQOMS/Orbitrap; Thermo Fisher Scientific, Waltham, MA, USA) adapted from the method of Barnaba et al. [56]. Chromatographic separation was performed using an ACCLAIM Vanquish PA 2 column (150 × 3 mm, 2.7 µm particle size). The mobile phase consisted of water/formic acid 100 mM/NH_4_HCO_2_ 20 mM 10% (A), acetonitrile 5% (B), and water 85% (C). Eluent A remains constant throughout the entire analytical run, while eluent B reaches 85% after 17 min, then returns to 5% to recondition the column. Total run time 21 min, with a flow rate of 0.4 mL/min. Mass spectrometric analysis was conducted with a Full MS scan—data dependent (MS/MS) experiment setting a resolution of 70,000 FWHM (*m*/*z* 200, 1.5 Hz) over a scan range of 200–2000 *m*/*z*. Raw material samples were prepared as for TPC analysis while extracted samples were appropriately diluted and filtered with a PTFE membrane. Phenolic compounds, shown in Table 2, were identified by comparison with authentic standards based on retention time, accurate mass (mass error < 5 ppm), and whenever possible, MS/MS fragmentation patterns. Linearity was confirmed in the range between 0.01 and 7 mg/L. Repeatability (calculated as RSD% on three replicates) of 10% and uncertainty (σ/√2) of 7%. Appendix A shows the LC-HRMS chromatograms of the apple pomace sample used in test REP4. Targeted analysis was performed using analytical standards (Merck Life Science, Milan, Italy) and solvent calibration possibly corrected with matrix addition, whereas compounds identified through suspect screening approach were quantified using the calibration curve of structurally similar compounds.

TDS: thermobalance, model MA 110.R (Radwag, Radom, Poland).

ORAC and TPC were found to be the most representative measures of the overall bioactivity of AP and APEs, as found for example by Kschonsek et al. in their study of the antioxidant capacity of 15 apple cultivars [57], while Zielińska et al. demonstrated a highly significant relationship between TPC, total flavonoid content, and the DPPH antioxidant essay [55].

### 2.4. Statistical Analysis

The statistical analysis was performed using XLSTAT version 2024.4 (Lumivero, Denver, CO, USA). The Kruskal–Wallis test with multiple pairwise comparisons using Dunn’s procedure was applied to evaluate significant differences among the different experiments. A Spearman correlation test (significance level alpha = 0.01) was conducted to assess the relationships between the investigated compounds, the cavitation passes and the temperature variables. Additionally, a Linear Regression (LR) model was developed to create a predictive framework for estimating ORAC and TPC yields based on temperature and cavitation passes.

## 3. Results

### 3.1. AP Biochemical Characterization

Table 3 shows the quantities measured for the AP samples used for the different extraction tests. The four most abundant individual phenolic compounds are shown, representing at least 70% of all individual phenolics. A large variability in TPC content (more than 100%) and ORAC levels emerged. AP used in REW1 (whole fruit) showed the lowest levels of ORAC and phlorizin content, significantly different from REP1 and REP2, respectively. AP used in tests REP1, REP2, REP3 and REP4 showed a more limited variability for ORAC and the content of phlorizin, with REP1 and REP4 showing the lowest content of epicatechin (REP1 significantly different from REP2), and REP4 also showing the lowest content of chlorogenic acid and procyanidin B2 (both significantly different from REP3) and total sugars (significantly different from REP2), possibly due to the prolonged preservation (until May, with the harvest season ending in fall) of the fruit used for REP4. While the lower content of sugars in the AP sample used in REP4 could suggest a certain degree of fermentation, possibly also associated with the degradation of bioactive compounds, interestingly, phlorizin appeared more resistant to degradation during apple preservation compared to other compounds. The content of total sugars was higher than 65% of dry weight, except for REP4 with a content lower than 50%. No significant associations arose among TPC, the considered individual phenols and ORAC.

Appendix A shows the concentration of all the compounds reported in Table 2 for AP samples.

### 3.2. APE Biochemical Characterization and Extraction Yield

#### 3.2.1. TPC and ORAC

Figure 2 shows the following quantities for each sample collected during the extraction tests, represented as a function of the number of passes of the entire volume through the cavitation zones (cavitation passes):Temperature;Cavitation number in the impeller (i) and Venturi-shaped reactor (v) zones, depicted as tags to the temperature curve;Extraction yield, computed as the ratio of TPC content or ORAC level in APEs to the corresponding levels in AP, normalized to the dry biomass;Peak process yield, depicted as tags to extraction yield data points and computed as the consumed energy (Wh) needed to obtain 1 mgCAT of TPC, or 1 mgTE of ORAC, from 1 g of dry AP. Hence, the process yield increases with the decrease of the computed quantity.

Cavitation passes were computed as the flow rate divided by the volume of the mixture, in turn assessed as the sum of the initial water volume of water and the water content of the biomass and multiplied by the time.

Process yield always peaked with the first sample, which was also associated with the highest or indistinguishable TPC or ORAC level. Thus, in tests REW1, REP1, REP2 and REP4, with all the biomass pitched in at the beginning of the process, peak process yield occurred 14, 22, 22 and 7 min, respectively, after the beginning of the process, or 10 min for tests REW1, REP1 and REP2 (60, 36 and 36 cavitation passes, respectively), and 2 min for test REP4 (10 cavitation passes) after the end of complete biomass insertion.

In REP1 and REP2, although the TPC extraction yield was quite similar until the end of the processes, with nearly 100% yield at the first sampling point, the ORAC yield of REP2 was significantly higher; moreover, both TPC and ORAC decreased abruptly at the end of the processes. In REW1, both TPC and ORAC yields were lower compared to the other tests at the beginning of the process, but later partially recovered (ORAC) or did not decrease further (TPC). In REP4, the initially lower TPC yield level measured only 2 min after the end of complete biomass insertion grew at the end of the process, yet likely the time of peak level was missed; however, the ORAC yield was quite high at the first sampling point and reached the highest level across all tests and the end of the process.

#### 3.2.2. Individual Phenolics and Total Sugars

Figure 3 shows the following quantities for each sample collected during the extraction tests, represented as a function of the cavitation passes:Temperature;Cavitation number in the impeller (i) and Venturi-shaped reactor (v) zones, depicted as tags to the temperature curve;Extraction yield, computed as the ratio of the content of individual phenolic compounds or total sugars in APEs to the corresponding levels in AP, normalized to the dry biomass;Peak process yield, depicted as tags to extraction yield data points and computed as the consumed energy (Wh) needed to obtain 1 mg of individual phenols from 1 kg of dry AP. Hence, the process yield increases with the decrease of the computed quantity.

Based on Figure 3a, the extraction yield of chlorogenic acid, which was the most abundant phenolic compound in AP (Table 2), was higher at the first sampling point and nearly 100% for all tests except REP4, where the yield was higher than 70% just 2 min after the end of complete biomass insertion and nearly reached 100% at the end of the process. The content of chlorogenic acid slightly decreases during the processes, except for a late partial recovery in REW1, which also showed an initially lower extraction yield. The process yield concerning chlorogenic acid always peaked at the first sampling point.

Based on Figure 3b, the extraction yield of phlorizin was stable or slightly decreasing during the process, except for test REP2, where it remarkably dropped after the first sampling point. It was also significantly higher in REW1 and REP1 than in REP3. The initial extraction yield for tests REW1, REP2 and REP3 was nearly 100%. Like chlorogenic acid, also the phlorizin extraction yield was higher than 70% in REP4 just 2 min after the end of complete biomass insertion and nearly reached 100% at the end of the process, thus not showing the sharp drop as in REP2. The process yield concerning phlorizin always peaked at the first sampling point.

Based on Figure 3c, the extraction yield of epicatechin was stable during the process, except for test REP2, where it dropped to very low levels after the first sampling point. It was also significantly higher in REW1 and REP1 than in REP3. The initial extraction yield for tests REW1 and REP1 was nearly 100%. Like chlorogenic acid and phlorizin, also the epicatechin extraction yield was higher than 70% in REP4 just 2 min after the end of complete biomass insertion and nearly reached 100% at the end of the process, thus not showing the sharp drop as in REP2. The process yield concerning epicatechin always peaked at the first sampling point, with the lower process yield for test REP1 due to the substantially lower content in the AP used in that test (Table 2).

Based on Figure 3d, the extraction yield of procyanidin B2 was stable during the processes of tests REW1 and REP3, while it significantly dropped in REP1 and increased in REP2 after the first sampling point. In REP2, the extraction yield increased to nearly 100% at the end of the process, while it was nearly 100% at the initial sampling point in REW1 and REP3. In REP4, the extraction yield of procyanidin B2 was higher than 70% just 2 min after the end of complete biomass insertion and nearly reached 100% at the end of the process. The process yield concerning procyanidin B2 always peaked at the first sampling point; however, it was quite stable during REP2 (1.8 to 2.7) due to the large increase in extraction yield.

Based on Figure 3e, the extraction yield of total sugars showed a slight decrease during the processes, a little steeper in test REP2. At the first sampling point, it was nearly 100% in tests REP1 and REP2, around 88% and 83% in REW1 and REP3, and higher than 90% in REP4, where it reached 100% at the end of the process.

Table 4 shows the quantities measured for the APE samples at the peak process yield, except for procyanidin B2 in test REP2, and all quantities in test REP4, where the levels measured in the last samples are reported. The individual phenolic compounds shown are the same as in Table 3. REP4 showed the highest levels of TPC and ORAC, significantly higher than REP1 and REW1, respectively, despite also showing the lowest levels of chlorogenic acid and procyanidin B2, significantly lower than all the other tests and REP3, respectively. In agreement with data shown in Table 3, REP4 also showed the lowest level of total sugars, significantly lower than REP2.

Appendix A shows the concentration of all the compounds reported in Table 2 for APE samples at the peak process yield.

### 3.3. Mass Extraction Yield and Estimated Composition of Dry Extracts

Table 5 shows the TDS levels measured for the last samples of tests REP2, REP3, and REP4, which represent the mass extraction yield relative to the unit dry biomass. The estimated contents of the dominant phenolic compounds and total sugars in the TDS are also shown, with total sugars largely dominating all potential dry extracts. Notably, the content of epicatechin is quite homogenous, while REP4 shows a higher content of phlorizin with a significant difference compared to REP2, and a lower content of chlorogenic acid and procyanidin B2, significantly compared to REP3.

## 4. Discussion

The high content of total sugars, always higher than 45% of the dry biomass (Table 3), 40% of APEs (Table 3) and 65% of TDS, i.e., potential dry extracts (Table 4), represents a constraint to the relative content of bioactive compounds. However, industrially generated apple by-products could contain substantially fewer total sugars, or sugars could be separated from the dry extracts, for example via membrane nanofiltration.

The results obtained in this study offer new insights into the feasibility and potential advantages of the use of HC processes in the extraction of apple by-products. Based on the data shown in Figure 2b, the ORAC yield observed in test REP1 (around 50 °C) was much lower than in test REP2 and REP4 (around 80 °C), under intermediate cavitation conditions, i.e., cavitation number in the Venturi reactor zone, between REP2 and REP4. The relatively high ORAC level shown by REP1 in Table 4 was due to the highest level in the corresponding AP, as shown in Table 3. Thus, for the sake of high ORAC yield, extraction processes should be carried out at temperatures substantially higher than 50 °C. This was most likely due to the action of the polyphenol oxidase enzyme, particularly abundant in apples, whose activity peaks between 25 and 35 °C and begins to degrade above 40 °C while maintaining a half-life of 12 min at 65 °C, and is inactivated at 80 °C [58].

Focusing on tests carried out around 80 °C and based on the peak extraction and process yields, HC extraction processes can be advantageously limited to no more than 30 to 50 passes of the entire volume of the mixture through the cavitation zones (approximately 10 to 20 min) after the end of complete biomass insertion. Indeed, based on Figure 2 and Figure 3, TPC, ORAC and the content of most individual compounds decreased or remained stable afterwards, with the notable exception of procyanidin B2, which looked like to be less sensitive to cavitation conditions and process time, while significantly affected by process temperature.

No dependence on the biomass to water ratio was shown, as TPC, ORAC, chlorogenic acid and phlorizin achieved their potential content in test REP2 with the highest ratio of dry biomass to water of 1:15 and, consequently, the maximum load of soluble extractives and viscosity in the processed mixture. It is known that, all else being equal, the pressure peak and energy of the shockwaves generated after cavitation bubble collapse decrease very fast with increasing viscosity at a certain distance from the bubble center and attenuate faster with the distance [59]. However, in the case of aqueous mixtures containing solid particles, all else being equal, the cavitation efficiency increases due to the creation of further cavitation nuclei, the increase in slip velocity and turbulent kinetic energy, and the decrease in the average size of solid particles due to cavitation-driven erosion [60]. In the considered extraction tests, the tendency to decreased cavitation intensity due to increasing viscosity was likely compensated by the cavitation intensification effect due to the increasing content and decreasing size of solid particles, i.e., the insoluble residues of the extraction process, hence the observed general independence of the extraction yield on the biomass content.

The extraction yield showed a compound-specific sensitivity to the details of the cavitation regime. Limited to the first sampling point, associated with the peak process yield, with the exception of the second sampling point for test REP4 and the last sampling point for procyanidin B2 in test REP2, Figure 3a shows that the short-term extraction yield of chlorogenic acid in test REW1 was the lowest, which could be associated with the highest levels of the cavitation number, thus lowest cavitation intensity, compared to the other tests. In contrast, phlorizin (Figure 3b) and procyanidin B2 (Figure 3d) appeared almost insensitive to the cavitation number. Epicatechin (Figure 3c) showed a dependence on the cavitation regime, with higher extraction yield for cavitation number in the Venturi reactor zone between 0.07 (REP4) and 0.11 (REW1), than 0.03 (REP2). Such contrasting behaviors could be ascribed either to the different degree of binding of each compound to complex polysaccharides such as cellulose, or the sensitivity to oxidation, which is known to be related to the HC-driven generation rate of hydroxyl (•OH) radicals, in turn generally increasing, in a Venturi-shaped HC reactor, with decreasing cavitation number [61].

ORAC, phlorizin and epicatechin also showed a remarkable sensitivity to the process time under high-intensity cavitation, showing fast drops after 50 to 100 passes (approximately 20 to 30 min) after the end of complete biomass insertion, such as in tests REP2 and REP3, with phlorizin appearing more sensitive to the cavitation number and epicatechin to the process time.

A statistical predictive analysis was performed using temperature and cavitation passes as predictors. The Spearman correlation test revealed no significant correlations between the cavitation passes and the extracted compounds, except for catechin-fisetinidol dimers (R^2^ = 0.624). Conversely, temperature exhibited inverse correlations with several extracted compounds, including procyanidin A1 (R^2^ = −0.736), procyanidin A2 (R^2^ = −0.692), procyanidin B1 (R^2^ = −0.616), and rutin (R^2^ = −0.663); however, all such compounds were present in very low quantities.

The LR analysis including both variables temperature and cavitation passes, revealed interesting predictive capacity for the yield of ORAC and TPC. Due to differences in the extraction process, the models were analyzed separately for the extracts of whole apple (test REW1) and AP (tests REP1 to REP4). Regarding REW1, the linear regression demonstrated a good explanatory capacity (R^2^ = 0.694 for ORAC and 0.66 for TPC), with a low root mean squared error (RMSE) equal to 3.100 and 3.573, for ORAC and TPC respectively, and an optimal Prediction Criterion (PC) equal to 0.919 and 1.019, respectively. In the case of the AP extract, the linear regression showed a moderate explanatory capacity for ORAC (R^2^ = 0.365) and a good explanatory capacity for TPC (R^2^ = 0.603), with acceptable RMSE levels of 11.14 and 10.15, respectively. The PC was optimal for ORAC (1.05) and acceptable for TPC (0.638).

Figure 4 shows the linear regression plots and the confidence intervals of the regression.

Equations (2) and (3) represent the predicted levels of ORAC and TPC, respectively, as a function of the temperature and the cavitation passes, for the whole apple extract (test REW1). Equations (4) and (5) have the same meaning for APEs (tests REP1 to REP4).(2)$$\mathrm{ORAC}\;=\;-103.919+1.979\;\times \;T+1.979\;\times {\;10}^{-2}\times CP,$$(3)$$\mathrm{TPC}\;=\;-56.287+1.447\;\times \;T+2.087\;\times {\;10}^{-2}\times CP,$$(4)$$\mathrm{ORAC}\;=\;41.230+0.601\;\times \;T\;-\;6.436\;\times \;10^{-2}\times CP,$$(5)TPC = 104.660 − 0.605 × T+0.171×CP
where *T* is the temperature (°C) and *CP* is the cavitation passes.

ORAC increased with process temperature in both extraction types, and increased with cavitation passes with the whole apple, but decreased with AP. Apparently, fewer cavitation passes are required for optimizing the extraction of AP towards the ORAC antioxidant activity, compared to whole apple, possibly due to the more complex structure of whole apple. Conversely, TPC increased with cavitation passes in both extraction types, and increased with temperature with the whole apple, but decreased with AP, with smaller dependence on temperature compared to ORAC.

It is likely that a longer exposure to cavitation helps extracting more phenolic compounds, but their degradation lowers the antioxidant activity; as well, higher process temperatures helped preserving the antioxidant activity of extracted phenolics, due to the inhibition of the polyphenol oxidase.

Overall, for the optimization of the extraction and process yields of TPC, ORAC and individual phenolic compounds, relative to the dry biomass, these results suggest that process temperatures should cautiously be not lower than 80 °C, cavitation passes should be no more than 30 to 40 (process times lower than 15–20 min) after the end of complete biomass insertion, and the cavitation number in the reactor zone should be as close as possible to 0.07.

This study has a few important limitations, which are listed below.

This was a retrospective study of experiments originally aimed at investigating the feasibility and general performance of the HC-based processing of apple by-products, thus the extraction tests, were carried out without a proper design of experiments and with different lots of apples, leading to a remarkable variability of AP composition, as shown in Table 3, and to levels and content of the different properties and compounds, either relative to the dry raw biomass or the TDS (Table 4 and Table 5, respectively), generally unrelated to the actual extraction yield. This was the reason why Figure 2 and Figure 3, showing the extraction yield, are more representative of the actual performance and sensitivity of the HC-based processes.In tests REP1, REP2, REP3 and REP4, AP was produced from the whole fruit using a pilot-scale hydropress. The use of industrial by-products could improve the standardization of AP, reduce the total sugars content and enhance the reproducibility of the results, which is planned as a subject of further research.The structure of the extracted phytocomplexes was not investigated. For example, HC-based extracts of red orange by-products were found to consist of stable phytocomplexes with flavonoids adsorbed onto the surface of pectin [45,62]. The mechanisms underlying the generation of pectin-polyphenols conjugates, using both citrus and apple commercial pectin, were identified, for example in the case of hydroxytyrosol, as the adsorption onto the surface of pectin, resulting in relatively weak non-covalent bonds, and the free radical method that produces stronger covalent bonds [63]. It can be hypothesized that HC processes intensify both conjugation mechanisms: adsorption, due to the greatly enhanced mass transfer rate produced by the HC-induced turbulence; likely more important, the formation of strong covalent bonds, which can be boosted due to the HC-based effective generation of hydroxyl radicals (•OH) [39,61]. However, the extracted red orange pectin showed a very low degree of esterification of 17.05% [62], while pectin extracted from Renetta variety apples showed a substantially higher degree of esterification of 74.2% [27], associated with a higher degree of hydrophobicity [64]. While a stable conjugation of apple polyphenols and pectin could improve metabolic and cardiovascular outcomes [17,18,28,65], whether our HC-based process could lead to such conjugation remains to be investigated and will be the subject of further research, including in vivo experiments.

While more structured experiments are needed to consolidate the presented findings, this retrospective study provided novel and potentially useful information about the optimization of the HC-based extraction processes.

### Scaled-Up Production of Dry Extracts

Dry water-soluble extracts from plant resources rich in bioactive compounds can represent high-value products for the food [66], nutraceutical [45], pharmaceutical and cosmeceutical [67] sectors. The dry form is convenient in the light of preservation, storage and transportation.

Based on a previous study [68], Figure 5 shows a general scheme of the production steps of dry extracts from plant-derived resources, such as AP.

Table 6 shows the main assumptions about the energy balance in the different steps of the production of dry extracts of AP, assuming a water volume of 1000 L.

Figure 6 shows the scenario of overall and specific (per unit mass of dry extract) energy consumption as a measure of process yield, assuming a fixed water volume of 1000 L, as a function of the water to biomass ratio (dry weight), for the typical AP moisture level of 83%, and process time of 20 min.

The overall energy consumption grows with decreasing water to dry AP ratio, while the specific energy follows an opposite trend. Spray drying, followed by vacuum drying, accounts for most of the energy consumption (around 65% and 24%, respectively), with the HC-based extraction step accounting for just 2.0 to 2.45%, due to its efficiency and the short process time required. It should be noted that since the water needs to be heated to 80 °C before the beginning of the extraction process, the energy consumption ascribed to the extraction step has been underestimated; however, the additional energy required could be minimized in a continuous production setting through energy recovery in a heat exchanger after centrifugation or vacuum drying. Finally, the decanter and centrifuge steps account for quite few percentage points of the overall energy consumption and their replacement with other separation systems, such as filter bag or filter press, should not significantly change the energy balance.

As the specific energy consumption halves with the water to dry AP ratio decreasing from 25:1 (20.5 kWh/kg of dry extract) to 10:1 (10.3 kWh/kg of dry extract), it is advisable to use as high an AP concentration as possible. Moreover, assuming phlorizin as the reference and most interesting compound, with a minimum effective dosage of 100 mg per day concerning human gut microbiota homeostasis [74], and its content in the dry extract around 1000 mg/kg as per Table 4, the specific energy consumption to obtain a dry extract containing 100 mg of phlorizin would decrease from about 2 kWh to 1 kWh with the water to dry AP ratio decreasing from 25:1 to 10:1. The mass of a potential dry extract containing 100 mg of phlorizin and including sugars would be around 100 g, decreasing to around 30 g after removing sugars.

These quantitative assessments may help drafting technical and economical feasibility plans for projects aimed at the production of dry extracts from AP. However, country- or economic zone-specific regulations could hinder either the adoption of the HC-based technology or the actual marketing of the derived products.

## 5. Conclusions

HC-based processes at the pilot scale afforded the integral extraction of macro- and micro-nutrients of AP up to nearly their potential levels in less than 20 min. It was found that the extraction should be carried out at temperatures not lower than 80 °C to avoid enzymatic oxidation and degradation of bioactive compounds. The extraction yield of bioactive compounds was found to be sensitive to the cavitation regime, with a cavitation number in the reactor zone around 0.07 delivering the highest yield.

A process chain leading to dry extracts as the end products was hypothesized, finding that the energy required to produce 30 to 100 g of dry extract (without or including sugars, respectively) containing 100 mg of phlorizin, an important bioactive dihydrochalcone characteristic to apples, using a reasonably high biomass to water ratio, was predicted around or lower than 1 kWh, with HC contributing for less than 2.5% to the overall energy consumption.

## Figures and Tables

**Figure 1 foods-14-01915-f001:**
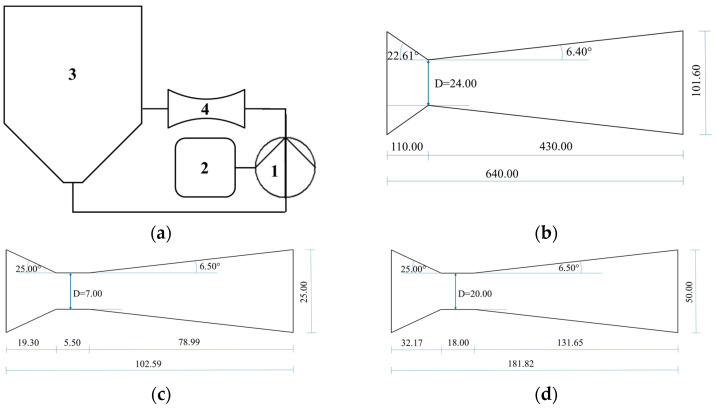
HC devices and reactors used in the experiments: (**a**) general layout, with numbers indicating 1—centrifugal pump; 2—electronic control panel with inverter (HC50 and HC300); 3—inline tank; 4—Venturi-shaped reactor; (**b**) reactor with throat area of 24 mm (HC200); (**c**) reactor with throat area of 7 mm (HC50); (**d**) reactor with throat area of 20 mm (HC300).

**Figure 2 foods-14-01915-f002:**
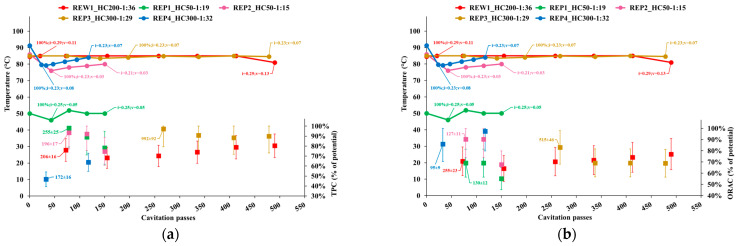
Temperature, cavitation number, target quantity as a percentage of its potential level and peak process yield for the samples collected in each test: (**a**) TPC; (**b**) ORAC.

**Figure 3 foods-14-01915-f003:**
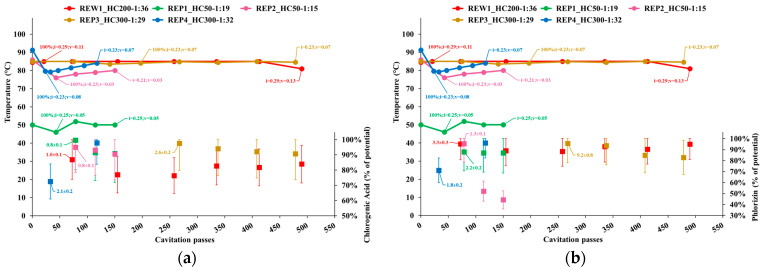
Temperature, cavitation number, target quantity as a percentage of its potential level and peak process yield for the samples collected in each test: (**a**) Chlorogenic acid; (**b**) Phlorizin; (**c**) Epicatechin; (**d**) Procyanidin B2; (**e**) Total sugars.

**Figure 4 foods-14-01915-f004:**
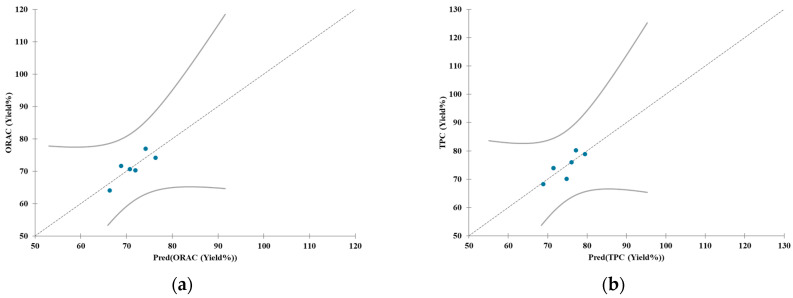
Linear regression plots. The relationship between the predicted yield (%) (X-axis) and the actual measured values (Y-axis) is represented by blue dots, while the curves indicate the confidence intervals of the regression: (**a**) ORAC for whole apple; (**b**) TPC for whole apple; (**c**) ORAC for AP; (**d**) TPC for AP.

**Figure 5 foods-14-01915-f005:**
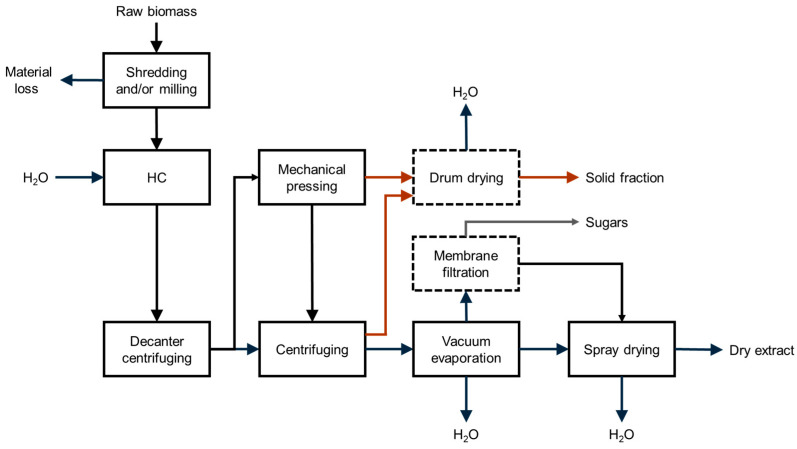
General scheme of the steps involved in the production of dry extracts from HC-based processing of AP. Black arrows refer to water–biomass mixture or water; brown arrows refer to insoluble wet residues; grey arrow refers to sugars. Dashed steps are not considered in the quantitative assessment.

**Figure 6 foods-14-01915-f006:**
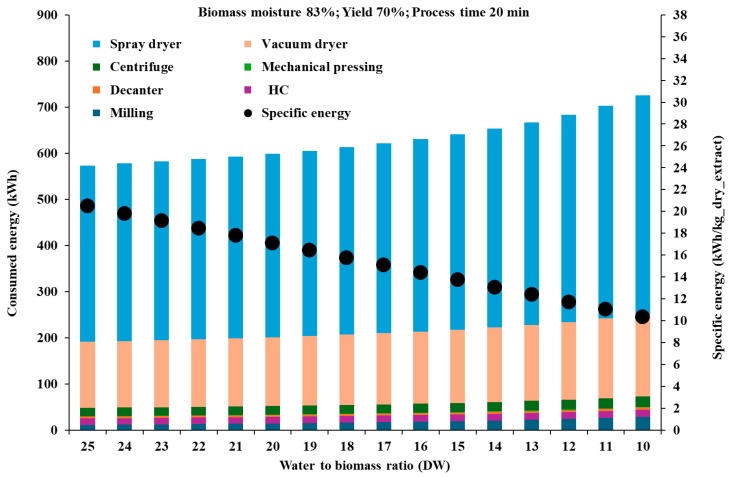
Scenario of the chain of process steps leading to the production of dry extracts from AP, with total energy consumption and energy consumption per unit mass of dry extract represented as a function of the water to dry biomass ratio, assuming biomass moisture of 83% and process time of 20 min.

**Table 1 foods-14-01915-t001:** Basic features of the extraction tests.

Test ID	Date	Biomass ^a^	Device	Fresh Biomass (kg)	Dry Biomass (kg)	Concentration (Dry Biomass to Water) ^b^	Time (min)	Temp. (°C)
REW1	October 2023	Whole	HC200	28.7	4.8	1:36	95	84.4 ± 1.4
REP1	December 2023	Pomace	HC50	11.4	2.1	1:19	42	49.6 ± 1.2
REP2	December 2023	Pomace	HC50	17.4	2.9	1:15	42	79.8 ± 3.8
REP3	January 2024	Pomace	HC300	48.9	8.4	1:29	102	84.5 ± 0.6
REP4	May 2024	Pomace	HC300	37.3	6.0	1:32	25	82.6 ± 4.2

^a^ Whole fruit or pomace obtained after juice squeezing. ^b^ The ratio includes the water contained in the fresh biomass.

**Table 2 foods-14-01915-t002:** Targeted and suspect screening phenolic compounds quantified and qualified with HPLC-HQOMS.

Compound	RT (min)	[M-H]^−1^ (*m/z*)	Fragments (*m/z*)	R^2^	C0 (Offset)	C1 (Slope)
Targeted analysis						
Caffeic acid	7.8	179.0350	135.044	0.999	−490,716	16,664,740
Catechin	7.0	289.0718	245.083; 109.029	0.995	58,088	324,723
Chlorogenic acid	7.1	353.0878	191.056	0.997	24,340	6,207,023
Cinnamic acid	11.9	147.0452	87.924	0.981	32,156	38,267
Epicatechin	7.4	289.0723	245.082; 109.029	0.997	85,095	3,277,577
Phloretin	14.9	273.0769	167.034; 119.049	0.997	2,664,383	49,980,513
Phlorizin	10.2	435.1297	273.077; 167.034	0.998	177,148	4,406,812
Procyanidin A1	9.4	575.1182	285.042; 125.025	0.997	−152,227	1,590,501
Procyanidin A2	9.9	575.1182	285.040; 125.025	0.997	−81,492	1,706,784
Procyanidin B1	6.7	577.1357	289.074; 125.023	1.000	−4476	1,458,512
Procyanidin B2	7.8	577.1357	289.070; 125.023	0.998	3857	1,549,692
Procyanidin B3	7.1	577.1357	289.072; 125.023	0.999	−76,337	1,835,583
Quercetin	14.2	301.0354	151.003; 178.999	0.999	109,125	11,965,356
Quercetin-3-glucosyde/Hyperoside	10.0	463.0882	300.029; 271.026	1.000	−845,470	14,229,433
Quercitrin	10.7	447.0933	300.030; 271.026	0.995	59,526	5,185,775
Quinic acid	7.3	191.0561	85.028	0.999	−18,145	137,859
Rutin	9.6	609.1461	300.030; 271.026	0.998	24,427	3,295,895
Suspect screening analysis						
Glucosyl-quinic acid	5.9	515.1617	-	Use Quinic acid		
Dehydrodicaffeoylquinic acids	9.0	513.1039	-	Use Caffeic acid		
Di-O-caffeoylquinic acid	10.7	515.1195	191.046; 135.046	Use Caffeic acid		
Trimers C-C-C	8.5	865.1985	125.025; 289.074	Use Catechin		
Dimers C-F	8.5	561.1402	289.074; 125.026	Use Catechin		

**Table 3 foods-14-01915-t003:** Measured properties of AP samples used in the extraction tests. Concentration levels are relative to the dry biomass.

Test ID	TPC (mgCAT/g_DW) ^a^	ORAC (mgTE/g_DW) ^b^	Phlorizin	Chlorogenic Acid	Epicatechin	Procyanidin B2	Total Sugars (mg/g_DW)
			(mg/kg_DW)				
REW1	9.8 ± 0.7 ^ab^	8.5 ± 0.8 ^b^	492 ± 34 ^b^	1712 ± 119 ^ab^	672 ± 47 ^ab^	918 ± 64 ^ab^	695 ± 31 ^ab^
REP1	4.9 ± 0.5 ^b^	13.7 ± 1.2 ^a^	628 ± 61 ^ab^	1628 ± 158 ^ab^	269 ± 26 ^b^	650 ± 63 ^ab^	697 ± 32 ^ab^
REP2	6.5 ± 0.6 ^ab^	10.4 ± 0.9 ^ab^	959 ± 82 ^a^	1627 ± 140 ^ab^	728 ± 43 ^a^	858 ± 74 ^ab^	738 ± 32 ^a^
REP3	5.8 ± 0.5 ^ab^	13.1 ± 1.2 ^ab^	640 ± 58 ^ab^	2209 ± 201 ^a^	640 ± 58 ^ab^	1298 ± 118 ^a^	646 ± 34 ^ab^
REP4	12.4 ± 0.8 ^a^	12.2 ± 1.1 ^ab^	798 ± 51 ^ab^	658 ± 42 ^b^	325 ± 21 ^ab^	314 ± 20 ^b^	466 ± 25 ^b^

^a^ CAT stands for (+)-catechin. ^b^ TE stands for Trolox equivalent. The letters ‘a’ and ‘b’ indicate statistically significant differences among the experiments, based on multiple pairwise comparisons using Dunn’s procedure.

**Table 4 foods-14-01915-t004:** Measured properties of APE samples at the peak process yield, except for procyanidin B2 in test REP2 and all quantities in test REP4, where the levels measured in the last samples are reported. Concentration levels are relative to the dry biomass.

Test ID	TPC (mgCAT/g_DW) ^a^	ORAC (mgTE/g_DW) ^b^	Phlorizin	Chlorogenic Acid	Epicatechin	Procyanidin B2	Total Sugars (mg/g_DW)
			(mg/kg_DW)				
REW1	7.4 ± 0.6 ^ab^	6.0 ± 0.5 ^b^	468 ± 38 ^b^	1488 ± 119 ^a^	677 ± 54 ^a^	846 ± 68 ^ab^	611 ± 35 ^ab^
REP1	4.8 ± 0.5 ^b^	9.4 ± 0.9 ^ab^	552 ± 54 ^ab^	1622 ± 159 ^a^	260 ± 25 ^b^	470 ± 46 ^ab^	665 ± 27 ^ab^
REP2	6.1 ± 0.5 ^ab^	9.4 ± 0.8 ^ab^	914 ± 82 ^a^	1547 ± 138 ^a^	543 ± 49 ^ab^	829 ± 85 ^ab^	712 ± 36 ^a^
REP3	5.6 ± 0.5 ^ab^	10.9 ± 1.0 ^ab^	612 ± 57 ^ab^	1474 ± 151 ^a^	348 ± 32 ^ab^	1254 ± 116 ^a^	535 ± 24 ^ab^
REP4	7.9 ± 0.6 ^a^	11.9 ± 1.1 ^a^	767 ± 52 ^ab^	644 ± 51 ^b^	289 ± 23 ^ab^	303 ± 24 ^b^	395 ± 22 ^b^

^a^ CAT stands for (+)-catechin. ^b^ TE stands for Trolox equivalent. The letters ‘a’ and ‘b’ indicate statistically significant differences among the experiments, based on multiple pairwise comparisons using Dunn’s procedure.

**Table 5 foods-14-01915-t005:** Available TDS of APE samples relative to the dry biomass and estimated content of individual phenols and total sugars.

Test ID	Passes ^a^	TDS (mg/g_DW)	Phlorizin	Chlorogenic Acid	Epicatechin	Procyanidin B2	Total Sugars (g/kg)
			(mg/kg)				
REP2	107	685 ± 3 ^b^	620 ± 64 ^b^	2153 ± 221 ^ab^	356 ± 37 ^a^	1211 ± 124 ^ab^	867 ± 45 ^a^
REP3	281	692 ± 3 ^ab^	766 ± 74 ^ab^	2896 ± 281 ^a^	456 ± 44 ^a^	1740 ± 170 ^a^	751 ± 33 ^ab^
REP4	84	720 ± 5 ^a^	1068 ± 85 ^a^	897 ± 71 ^b^	403 ± 32 ^a^	422 ± 33 ^b^	656 ± 30 ^b^

^a^ Cavitation passes after the end of complete biomass insertion. The letters ‘a’ and ‘b’ indicate statistically significant differences among the experiments, based on multiple pairwise comparisons using Dunn’s procedure.

**Table 6 foods-14-01915-t006:** Main assumptions for the energy balance in the production steps of dry extracts of AP, assuming a water volume of 1000 L.

Step	Quantity	Level	Unit	Source/Notes
Biomass	Specific heat of the dry biomass	1370	J/kgK	[69]
Milling	Material loss	0	%	data
	Specific energy consumption ^a^	50	kWh/ton	Personal experience with commercial bio-shredder
HC	Process time	20	minutes	Evidence from this study
	Temperature ramp	Constant at 80 °C		Evidence from this study
	Energy consumption per unit time	0.8	kWh/min	Based on test REP4 ^b^
	Yield of dry extract relative to dry biomass	500	g/kg_DW	Based on test REP4 (Table 4)
Decanter centrifuging	Separation efficiency	95%		[70]
	Moisture in separated material	75%		
	Specific energy consumption ^c^	3.38	kWh/ton	
Mechanical pressing	Specific energy consumption ^d^	30	kWh/ton	Personal experience
	Moisture in separated material	40%		
Centrifuge	Energy consumption per unit mass of water	15	kWh/ton	[71]
	Separation rate	100	%	Negligible errors due to 95% separation by the decanter
Vacuum dryer	Water evaporation rate	80%		[72]
	Energy consumption per unit mass of extract	150	kWh/ton	
Spray dryer	Energy consumption per unit mass of extract	1600	kWh/ton	[73]

^a^ Energy consumption per unit mass of raw material (FW). ^b^ In test REP4, 0.13 kWh/min for a water volume of 160 L. ^c^ Energy consumption per unit mass of extract. ^d^ Energy consumption per unit mass of separated material (FW).

## Data Availability

The data set and any information regarding the design and operation of the HC devices are available upon reasonable request from the corresponding author.

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
