# Peer review of "Sustainable Exploitation of Apple By-Products: A Retrospective Analysis of Pilot-Scale Extraction Tests Using Hydrodynamic Cavitation"

_foods, 2025, doi:10.3390/foods14111915_

Round 1
Reviewer 1 Report (Previous Reviewer 1)
Comments and Suggestions for Authors
Dear Authors,
I recommend to accept your paper in present form.
Kind regards,
Author Response
Thank you very much for your appreciation of our manuscript!

Reviewer 2 Report (New Reviewer)
Comments and Suggestions for Authors
Dear, the manuscript "Sustainable Exploitation of Apple By-Products: A Retrospective Analysis of Pilot-Scale Extraction Tests Using Hydrodynamic Cavitation" is quite interesting and worth investigation. It was improved significantly, when compared to previous versions. I agree with academic editor. Complementary analysis should be carried out or a deeper discussion and/or analysis should be done
Regards
Author Response
Please see the attachment.

Reviewer 3 Report (New Reviewer)
Comments and Suggestions for Authors
The current article proposes a sustainable exploitation of apple by-products by extraction of valuable compound trough hydrodynamic cavitation. Moreover, the study carried out with large quantities of raw material, being intended for industrial application. Overall is a complex study, that explore the best conditions for extraction by hydrodynamic cavitation.
However, there are some weaknesses related to the results presentation:
- The Results and Discussion Sections of this manuscript are difficult to read and understand.
- Some phrases are too long.
- The figures 2 and 3 do not show properly the data, they are difficult to understand and interpret.
- The manner that results are presented does not emphasize the best extraction condition.
Specific comments:
- Line 139: ρ should be the liquid density.
- Line 259: Explanation of the AAPH abbreviation is missing.
- Line 283-284: The composition of mobile phase is not clear. Explanation of the FA abbreviation is missing.
Over 65% of the references are in the last 5 years. Self-citation about18 %. Some references are in the wrong format.
Round 2
Reviewer 3 Report (New Reviewer)
Comments and Suggestions for Authors
The manuscript has been significantly improved. The results are presented and interpreted more clearly and in more detail.
Author Response
Thank you very much for the kind appreciation of our manuscript!
This manuscript is a resubmission of an earlier submission. The following is a list of the peer review reports and author responses from that submission.
Round 1
Reviewer 1 Report
Comments and Suggestions for Authors
Dear Authors,
your paper is interesting but I have some remarks and questions to you.
- There is no description of statistical analysis in the "Materials and Methods" section. Was it performed at all?
- The data presented in the "Results" section, neither in the tables nor in the text itself, have been statistically analyzed. In the absence of mathematical statistical analysis of your results, this paper can not be published in the present form.
- You use only ORAC assay for determination of antioxidant activity of tested extracts. Herbal extracts are multicomponent matrices with antioxidant activity determined by the set of different mechanism reactions, so antioxidant effect cannot be adequately tested using only one method. For these reasons, it is strongly recommended to use at least two different methods for determination of antioxidant activity in herbal extracts. I strongly recommend you to use different assays for determination of antioxidant activity in vitro of tested extracts. I recommend to use ABTS, DPPH or TFPH assays for determination of antiradical activity in vitro and CUPRAC or FRAP assays for determination of reduction activity in vitro.
- Why you did not determine quercetin glycosides in tested extracts? Quercetin glycosides are strong antioxidants so I strongly recommend to determine these compounds in apple extracts.
- Why did you express the total phenolic content (TPC) in the (+)-catechin equivalent? In my personal opinion it would be more appropriate to express TPC in gallic acid equivalent. These expression would help you to compare your results to data of other scientists.
Kind regards,
Author Response
Please see the attachment.
Response to Reviewer 1 Comments |
Dear Authors, your paper is interesting but I have some remarks and questions to you. |
Thank you very much for taking the time to review this manuscript and for the kind appreciation of our study. Please find below a point-by-point response to all your comments. |
Comment 1: There is no description of statistical analysis in the "Materials and Methods" section. Was it performed at all? |
Response 1: We are indebted to the esteemed Reviewer for helping us to substantially improve our manuscript. In Materials and Methods, we have added Section 2.5 “Statistical analysis”. |
Comment 2: The data presented in the "Results" section, neither in the tables nor in the text itself, have been statistically analyzed. In the absence of mathematical statistical analysis of your results, this paper can not be published in the present form. |
Response 2: A statistical analysis was performed and added to the Discussion section, including Figure 4. We thank the esteemed Reviewer for the help to improve our manuscript. |
Comment 3: You use only ORAC assay for determination of antioxidant activity of tested extracts. Herbal extracts are multicomponent matrices with antioxidant activity determined by the set of different mechanism reactions, so antioxidant effect cannot be adequately tested using only one method. For these reasons, it is strongly recommended to use at least two different methods for determination of antioxidant activity in herbal extracts. I strongly recommend you to use different assays for determination of antioxidant activity in vitro of tested extracts. I recommend to use ABTS, DPPH or TFPH assays for determination of antiradical activity in vitro and CUPRAC or FRAP assays for determination of reduction activity in vitro. |
Response 3: Thank you for this important comment. After a careful review of the specific literature on the antioxidant activity of apple and its by-products, we concludes that ORAC and TPC are likely the most effective measures. This approach has already been adopted by Kschonsek et al. in their study of the antioxidant capacity of 15 apple cultivars, and by Candrawinata et al. in their investigation of how clarification impacts polyphenolic compounds and antioxidant activity in commercial apple juices. Additionally, ZieliÅ„ska D. and Turemko M. demonstrated a highly significant relationship between TPC and DPPH (and TFC) for apple flesh extracts from 11 apple cultivars, a finding similarly observed by Panzela et al., who found a strong correlation between total polyphenols and DPPH. 1. ZieliÅ„ska D. and Turemko M. Electroactive Phenolic Contributors and Antioxidant Capacity of Flesh and Peel of 11 Apple Cultivars Measured by Cyclic Voltammetry and HPLC-DAD-MS/MS. Antioxidants (Basel). 2020(11):1054. doi: 10.3390/antiox9111054. PMID: 33126563; PMCID: PMC7694104. 2. Lucia Panzella, Milena Petriccione, Pietro Rega, Marco Scortichini, Alessandra Napolitano, A reappraisal of traditional apple cultivars from Southern Italy as a rich source of phenols with superior antioxidant activity, Food Chemistry, Volume 140, Issue 4, 2013, 672-679, ISSN 0308-8146, https://doi.org/10.1016/j.foodchem.2013.02.121. 3. Candrawinata, Vincent & Blades, Barbara & Golding, John & Stathopoulos, Costas & Roach, Paul. (2012). Effect of clarification on the polyphenolic compound content and antioxidant activity of commercial apple juices. International Food Research Journal. 19. 1055-1061. 4. Kschonsek J, Wolfram T, Stöckl A, Böhm V. Polyphenolic Compounds Analysis of Old and New Apple Cultivars and Contribution of Polyphenolic Profile to the In Vitro Antioxidant Capacity. Antioxidants (Basel). 2018 Jan 24;7(1):20. doi: 10.3390/antiox7010020. PMID: 29364189; PMCID: PMC5789330. The following text has been added at the end of Section 2.4: “ORAC and TPC were found to be the most representative measures of the overall bioactivity of AP and APE, as found for example by Kschonsek et al. in their study of the antioxidant capacity of 15 apple cultivars [reference: doi: 10.3390/antiox7010020], while ZieliÅ„ska et al. demonstrated a highly significant relationship between TPC, total flavonoid content and the DPPH antioxidant essay [reference: doi:10.3390/antiox9111054]”. |
Comment 4: Why you did not determine quercetin glycosides in tested extracts? Quercetin glycosides are strong antioxidants so I strongly recommend to determine these compounds in apple extracts. |
Response 4: We performed the evaluation of quercetin and quercetin glycosides on all the apple extracts, but we found low levels of both. For quercetin, we found a quantification lower than the detection threshold and for its glycosides (isoquercetin) we found about 5 mg/100 g of apple pomace dry mass. |
Comment 5: Why did you express the total phenolic content (TPC) in the (+)-catechin equivalent? In my personal opinion it would be more appropriate to express TPC in gallic acid equivalent. These expression would help you to compare your results to data of other scientists. |
Response 5: Thank you for your careful comment. We used that expression following a recent analysis of antioxidant activity of several apple cultivars. We have added the following text to the TPC point in Section 2.4: “TPC was expressed in (+)-catechin equivalent as recently used for the antioxidant capacity of flesh and peel of several apple cultivars [reference: doi:10.3390/antiox9111054].”. |
Response to Comments on the Quality of English Language |
Point 1: The English is fine and does not require any improvement. |
Response 1: Thank you for your appreciation. |

Reviewer 2 Report
Comments and Suggestions for Authors
The presented paper is a complex pilot study on the extraction procedures of apple by-products. It is well-written, with clear and appropriate graphical presentation. The study explores the utilization of apple by-products (AP), including defective fruits and pomace, which are typically discarded due to the lack of scalable extraction techniques. The research focuses on hydrodynamic cavitation (HC) as a method for extracting bioactive compounds using only water as a solvent. Phlorizin, a key bioactive compound almost exclusive to apples, is highlighted for its potential health benefits against chronic diseases. The study identifies critical process parameters, such as the necessity of high temperatures (>80°C) to deactivate polyphenol oxidase and a narrow cavitation number (~0.07) for optimal extraction. The reported extraction process achieves substantial macro- and micronutrient recovery within 20 minutes, regardless of biomass concentration (up to 50%). The predicted energy requirement for producing 30–100 g of dry extract containing 100 mg of phlorizin is approximately 1 kWh, with HC contributing minimally (<2.5%) to overall energy consumption. While the findings suggest an efficient extraction method, further validation regarding scalability and economic feasibility is necessary.
Questions to be Addressed in the Revised Manuscript:
- What are the main limitations of hydrodynamic cavitation compared to other extraction methods for apple by-products?
- How does the composition and quality of the extracted bioactive compounds compare to those obtained through conventional solvent-based extraction?
- Were any degradation products formed during high-temperature processing, and how do they impact the overall bioactivity of the extract?
- How does the proposed method perform when applied to different apple varieties or varying levels of ripeness?
- What are the estimated costs and feasibility of scaling up this process for industrial applications?
- Could the extracted compounds have potential applications beyond the food and pharmaceutical industries, such as in cosmetics or biodegradable packaging?
- How does the water-to-biomass ratio affect the yield and purity of the extracted compounds?
- Are there any regulatory concerns regarding the adoption of this extraction technique for commercial use?
Author Response
Please see the attachment.
Response to Reviewer 2 Comments |
The presented paper is a complex pilot study on the extraction procedures of apple by-products. It is well-written, with clear and appropriate graphical presentation. The study explores the utilization of apple by-products (AP), including defective fruits and pomace, which are typically discarded due to the lack of scalable extraction techniques. The research focuses on hydrodynamic cavitation (HC) as a method for extracting bioactive compounds using only water as a solvent. Phlorizin, a key bioactive compound almost exclusive to apples, is highlighted for its potential health benefits against chronic diseases. The study identifies critical process parameters, such as the necessity of high temperatures (>80°C) to deactivate polyphenol oxidase and a narrow cavitation number (~0.07) for optimal extraction. The reported extraction process achieves substantial macro- and micronutrient recovery within 20 minutes, regardless of biomass concentration (up to 50%). The predicted energy requirement for producing 30–100 g of dry extract containing 100 mg of phlorizin is approximately 1 kWh, with HC contributing minimally (<2.5%) to overall energy consumption. While the findings suggest an efficient extraction method, further validation regarding scalability and economic feasibility is necessary. |
Thank you very much for taking the time to carefully review this manuscript and the kind appreciation of our study. Please find below a point-by-point response to all your comments. |
Comment 1: What are the main limitations of hydrodynamic cavitation compared to other extraction methods for apple by-products? |
Response 1: Information about other extraction methods for apple by-products had been reported from line 112 (“Conventional extraction techniques of phenolic compounds from whole apples and AP include Soxhlet...”) to line 127. There was only one reported application of a hydrodynamic cavitation (HC) technique to the AP extraction, yet AP had been previously fermented – this information has been added to the text. Moreover, the following consideration was added: “However, a stator-rotor setup was used, which jeopardizes full scale applications due to the excessive energy cost of rotational HC reactors compared to static ones, such as Venturi or orifice constrictions [reference: doi:10.1016/j.trac.2024.117682]”. Therefore, this can be considered the first time for the application of HC comprising a linear static reactor to the extraction of AP, as well as further comparison with other methods is hardly possible at the present time. The results achieved in this study, which could represent a benchmark for competitive techniques, showed a close reproducibility of the results obtained under similar geometrical features and the same processing conditions (temperature, cavitation number and cavitation passes), using volumes of 50, 200 and 300 L, with the 300 L process showing process yields (extraction yield per unit consumed energy) comparable or even higher than the smaller devices. |
Comment 2: How does the composition and quality of the extracted bioactive compounds compare to those obtained through conventional solvent-based extraction? |
Response 2: As shown in Section 3.2, hydrodynamic cavitation afforded practically full extraction of bioactive compounds. That is, all the bioactive compounds extracted from the raw material through conventional solvent-based extraction (Section 2.4 “Analysis of Raw Material and Extracts”) have eventually and quickly been extracted using hydrodynamic cavitation. Thus, the composition of the HC-based extracts in terms of bioactive compounds did not differ from the extracts obtained using solvent-based extraction. In Section 4 (Discussion), it was further hypothesized that the obtained extract could contain phytocomplexes with stable conjugation of apple polyphenols and pectin, which would represent a distinct feature and possibly a functional advantage for the obtained extract but should be investigated in further research. |
Comment 3: Were any degradation products formed during high-temperature processing, and how do they impact the overall bioactivity of the extract? |
Response 3: Thank you for this important comment. Based on the results presented in Section 3.2 and discussed in Section 4, no obvious degradation occurred with increasing process temperature. On the contrary, both the content of individual phenolic compounds and overall bioactivity (measured as ORAC) were higher with processing temperature around 80°C compared to 50°C, while at 80°C the amount of certain individual compounds in the extracts, such as phlorizin and epicatechin, and the ORAC, appeared to depend on the cavitation regime represented by the cavitation number. We cannot rule out the formation of degradation products, such as from Maillard reactions, which we did not measure, but no obvious effects of high process temperature on individual bioactive compounds or overall bioactivity was observed. |
Comment 4: How does the proposed method perform when applied to different apple varieties or varying levels of ripeness? |
Response 4. Thank you for this insightful comment. As per Table 1, the tests were performed using batches of apples of the Renetta variety at different moments of the year. The apples were stored at the supplier facilities, and it is not possible in this retrospective study to know whether those batches were characterized by different levels of ripeness. Based on the presented data, we can only observe that there was no obvious dependence on the specific apple batches used in the experiments. Concerning apple varieties, we performed a few preliminary trials with the Golden variety, finding practically the same results obtained with the Renetta variety in terms of potential extraction. However, the quality of experiments and analysis performed with the Golden variety was insufficient for inclusion in this study. The motivation for the use of the Renetta variety was the substantially higher content of bioactive compounds compared to other varieties cultivated in the same geographic area (reference: doi:10.1021/jf049317z). This information was added to the last paragraph of the Introduction. Finally, we think that there is no reason why hydrodynamic cavitation should perform differently with any other apple variety, as the proposed method does not use chemicals or enzymes that could interact differently with different varieties, as well as the same method showed comparable performance in the application to completely different plant resources, among which orange by-products (doi: 10.3390/biomedicines13030686), spruce bark (doi: 10.1016/j.seppur.2024.130925), chestnut wood by-products (doi: 10.1007/978-3-031-38274-1_36), and whole almond kernels (doi: 10.3390/foods12050935). |
Comment 5: What are the estimated costs and feasibility of scaling up this process for industrial applications? |
Response 5: About the feasibility of scaling up this process, the present retrospective analysis of pilot-scale extraction tests of apple by-products using a Venturi-shaped reactor with linear flow showed a close reproducibility of the results obtained under similar geometrical features and the same processing conditions (temperature, cavitation number and cavitation passes), using volumes of 50, 200 and 300 L, with the 300 L process showing process yields (extraction yield per unit consumed energy) comparable or even higher than the smaller devices. This suggests a straightforward scalability. Moreover, due to the very fast extraction process and the high content of processed biomass, even a 300 L device could match the needs of an industrial process. About the costs, we did not elaborate, because they are far too dependent on the local cost of raw materials and utilities. While we think that the cost will be much less than proportional to the size, further elaboration was outside the scope of this study. |
Comment 6: Could the extracted compounds have potential applications beyond the food and pharmaceutical industries, such as in cosmetics or biodegradable packaging? |
Response 6: We think that the extracted compounds could have potential applications at least in cosmetics. About biodegradable packaging, not only the extracts but the insoluble residues after the extraction, especially cellulose, could find interesting applications. In this study, we focused on the analysis of the extraction processes; the analysis of potential applications, including more advanced functionality of the extracts, will be the subject of already planned research. |
Comment 7: How does the water-to-biomass ratio affect the yield and purity of the extracted compounds? |
Response 7: In the fourth paragraph of the Discussion it is stated that “No dependence on the biomass to water ratio was shown, as TPC, ORAC, chlorogenic acid and phlorizin achieved their potential content in test REP2 with the highest ratio of dry biomass to water of 1:15 and, consequently, the maximum load of soluble extractives and viscosity in the processed mixture”, followed by a further elaboration. At least within the concentration limits explored in our experiments, the water-to-biomass ratio did not appear to affect the yield and composition of the extracts. |
Comment 8: Are there any regulatory concerns regarding the adoption of this extraction technique for commercial use? |
Response 8: Thank you for this comment, that highlights an important point. The aspect of product regulation and related consumer safety is certainly a relevant issue and is particularly dependent on local regulations at the level of single countries or group of countries. In our experience, sometimes there is even some discretion in applying the regulation, including in the event of modifications to traditional processing methods. This is also why we think that delving deeper into such important topics is outside the scope of our study. However, we have added the regulatory issues to the barriers challenging industrialization at the end of Section 4.1: “However, country- or economic zone-specific regulations could hinder either the adoption of the HC-based technology or the actual marketing of the derived products.”. |
Response to Comments on the Quality of English Language |
Point 1: The English is fine and does not require any improvement. |
Response 1: Thank you for your appreciation. |

Round 2
Reviewer 1 Report
Comments and Suggestions for Authors
Dear authors,
thank you for your answers very much. Now this paper is really better than previously. It is very good that you include the description of your statistical analysis in the section "2. Materials and Methods" as I wrote before. However no statistical analysis elements are presented in the tables 2-4. Did you use ANOVA with post hoc tests or non-parametric criteria to assess the statistical significance of the obtained results? Without performing this analysis, it cannot be stated that the obtained results are statistically significant.
Kind regards,